# MULTI-TASK LEARNING WITH HYPERNETWORKS AND TASK METADATA

## ABSTRACT

Multi-task learning architectures model multiple related tasks simultaneously by sharing parameters across networks to exploit shared knowledge and improve performance. Designing multi-task architectures is challenging due to the trade-off between parameter efficiency and the ability to flexibly model task differences at all network layers. We propose Multi-Task Hypernetworks, a novel multi-task learning architecture which circumvents this trade-off, generating flexible task networks with a minimal number of parameters per task. Our approach uses a hypernetwork to generate different network weights for each task from small task-specific embeddings and enable abstract knowledge transfer between tasks. Our approach stands out from existing multi-task learning architectures by providing the added capability to leverage task-level metadata to explicitly learn task relationships and task functions. We show empirically that Multi-Task Hypernetworks outperform many state-of-the-art multi-task learning architectures on small tabular data problems, and leverage metadata more effectively than existing methods.

## 1 INTRODUCTION

Multi-Task Learning (MTL) is a machine learning paradigm where a set of related tasks are learnt jointly to transfer knowledge between them and improve generalisation performance of all tasks. One of the key challenges of MTL is designing network architectures to effectively transfer knowledge between tasks. Architectures can be broadly categorised into hard parameter sharing and soft parameter sharing (Ruder, 2017). Hard parameter sharing architectures transfer knowledge by sharing most network parameters between all tasks. These approaches are parameter-efficient as only a small set of model weights are learnt separately for each task, typically in deep layers (Caruana, 1997; Shui et al., 2019; Pascal et al., 2021). However, the flexibility of these models is limited as all tasks use an identical feature representation which may not be optimal (Guo et al., 2019; Ruder et al., 2019). Soft parameter sharing approaches instead learn unique weights for each task network, constrained by some mechanism such as regularisation to transfer knowledge between tasks. This enables the learning of *flexible* task networks, where distinct task-specific representations are learnt at all layers of the network. However, these techniques are not parameter-efficient as they require learning an entire set of model weights per task (Misra et al., 2016; Ruder et al., 2019; Yang & Hospedales, 2017). This highlights a trade-off in multi-task learning architectures between parameter-efficient task scaling and learning flexible task networks.

To address this trade-off, we propose *Multi-Task Hypernetworks*, a novel MTL architecture which learns flexible task networks with substantially fewer parameters per task than other soft parameter sharing approaches. We achieve this by learning a unique low-dimensional embedding vector of each task, which captures task relationships. Despite using task embeddings as the only task-specific parameters, our architecture can generate flexible task networks and enable abstract knowledge transfer. In our experiments, we use only ten parameters per task.

Our approach accomplishes this by utilising hypernetworks (Ha et al., 2017) for soft parameter knowledge transfer across tasks. Notably, this represents the first attempt within the research area to explore this methodology. Hypernetworks are neural networks that generate the weights of another neural network. Instead of directly learning the weights of a "target" network, which models the task of interest, hypernetworks produce them dynamically based on some input. Our proposed Multi-Task Hypernetwork uses a hypernetwork to generate a different set of weights for multiple

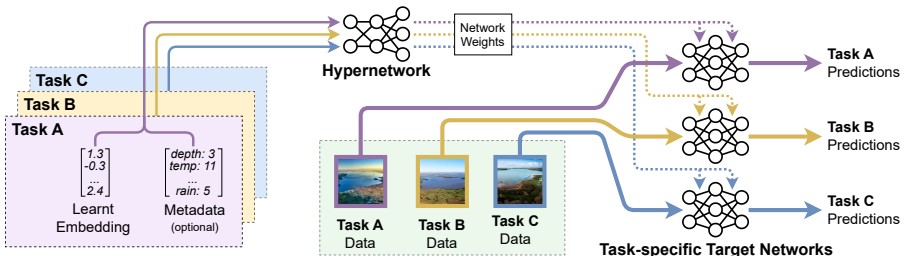

Figure 1: Multi-Task Hypernetwork conceptual diagram.

target networks which each model a different task. Network weights for all target networks are generated by the same hypernetwork, but use task-specific embeddings as input to the hypernetwork to generate entirely distinct weights (Figure 1). The use of a shared deep hypernetwork enables abstract knowledge transfer between tasks, while the task embeddings capture task relationships. This is a conceptual departure from previous soft parameter sharing approaches which share knowledge at the parameter level. We additionally show experimentally that the task embeddings learn "meaningful" task representations, in that they are predictive of task-level knowledge.

Another key benefit of Multi-Task Hypernetworks is that they can naturally integrate task-level information into the learning process. In some MTL applications, auxiliary data describing the tasks themselves is available. We refer to this as *metadata*, and argue that this complementary data source may provide valuable knowledge of task relationships. For example, consider the application of water quality remote sensing (Pahlevan et al., 2020; Graffeuille et al., 2022). The chlorophyll-$a$ concentration is estimated from water colour in multiple lakes, with each lake modelled as a different task. Data is typically scarce due to the high cost of data collection, thus, learning these tasks jointly can improve performance. In this application, some lake features such as depth, temperature and rainfall are known. These features are informative, as lakes with similar features are likely to have similar environmental mechanisms (Yang et al., 2022). When modelling each lake as a different task, these task-level features, which we call metadata, are informative with regards to task relatedness.

Beyond parameter-efficient generation of flexible task networks, Multi-Task Hypernetworks are also the first general MTL approach to learn task relationships and task functions explicitly from task-level metadata, rather than implicitly by jointly optimising task performances. This is achieved by using task metadata as input to the hypernetwork along with the task embeddings, such that target network weights are a function of their metadata. When metadata is available, leveraging it during training may help the model learn task relationships and when to transfer across tasks, a key challenge in MTL (Zamir et al., 2018; Fifty et al., 2021). This may be particularly important for small data problems where data is insufficient to accurately learn task relationships (Zheng et al., 2019). Existing MTL algorithms are designed to learn from data at the sample level, but not from data at the task level, and so cannot leverage metadata directly.

Experiments on a novel metadata-driven synthetic MTL dataset and three small tabular MTL datasets with task metadata demonstrate (1) our approach outperforms state-of-the-art MTL architectures on tabular data without metadata (2) including metadata substantially improves the performance of our approach (3) other existing approaches cannot effectively leverage this metadata.

For reproducibility, we make our code, datasets and synthetic dataset generator available online [1]. Section 2 proposes Multi-Task Hypernetworks. Section 3 covers experimental results. Section 4 explores related works in MTL and hypernetworks. Section 5 concludes the paper.

## 2 MULTI-TASK HYPERNETWORK

**Problem formulation.** In MTL, we consider a set of $T$ related tasks $\{\mathcal{T}_t\}_{t=1}^T = \{(\mathcal{D}_t, f_t)\}_{t=1}^T$, such that each task $\mathcal{T}_t$ has a deep learning function $\{f_t : \mathcal{X} \mapsto \mathcal{Y}\}_{t=1}^T$ and a set of training data $\{\mathcal{D}_t\}_{t=1}^T$ containing $n_t$ training instances $\mathcal{D}_t = \{x_i^t, y_i^t\}_{i=1}^{n_t}$ where $x_i^t \in \mathcal{X}, y_i^t \in \mathcal{Y}$. The goal of MTL is to jointly learn each task function $\{f_t\}_{t=1}^T$ from the training data of all tasks $\{\mathcal{D}_t\}_{t=1}^T$.

---

[1] https://anonymous.4open.science/r/Multi-Task-Hypernetworks-922C

**Metadata.** Metadata is data that describes and gives information about other data (Zheng et al., 2019). In the context of MTL, we define metadata as task-level data, that is, data describing the tasks themselves. For example, lake attributes in a problem where each lake is associated with a different task, or mechanical information about robot arms when modelling each arm's movements as a task. Metadata may come from auxiliary datasets or domain knowledge. We note that for some MTL applications, metadata is unavailable or undefined. Consider the learning of semantic segmentation and surface normal of an image as a multi-task problem (Misra et al., 2016); no relationship between these two tasks exists which can be expressed with task-level features. We observe that many tabular datasets and databases contain task-level metadata as shown in Section 3, but that such metadata is less common in vision problems, and hence concentrate our work on tabular data.

**Problem formulation with metadata.** When metadata is available, each task $\mathcal{T}_t$ additionally has an associated metadata feature vector $m_t \in \mathbb{R}^{d_m}$, such that $\mathcal{T}_t = (\mathcal{D}_t, m_t, f_t)$. The goal of MTL with metadata is to jointly learn the task functions $\{f_t\}_{t=1}^T$ from the task training data $\{\mathcal{D}_t\}_{t=1}^T$ and metadata $\{m_t\}_{t=1}^T$. When metadata is unavailable, we use the original MTL problem formulation.

Current MTL approaches are not designed to effectively leverage metadata. A naive approach to learn from metadata with existing techniques would be to append it as supplementary features to training data $\{\hat{x}_i^t\}_{i=1}^{n_t} = \{(x_i^t, m_t)\}_{i=1}^{n_t}$. This may not be an effective way to exploit this data, considering that these appended features would be constant for all data instances within a task.

**Method overview.** The foundation of our multi-task learning approach is a hypernetwork (Figure 1). The network weights $\theta_t$ of target function $f_t$ of task $t$ are not trained directly. Instead, they are generated by a shared hypernetwork $h$ according to $\theta_t = h(e_t)$, where $e_t$ is a $d_e$ dimensional embedding vector associated with task $t$. The hypernetwork generates the weights for each task's target network. Using task-specific embeddings allows the hypernetwork to generate different weights for each target network. Task embeddings are low-dimensional representations of target functions learnt by the Multi-Task Hypernetwork, which allow our model to learn task similarities and relationships.

Multi-Task Hypernetworks are parameterised by the weights $\theta_h$ that define $h$, and the $T$ task-specific embedding vectors $\{e_t\}_{t=1}^T$. Most model parameters are shared between all tasks since $|\theta_h| \gg Td_e$. However, our approach is best classified as a soft parameter sharing approach, given that each target network is generated with flexible weights at all network layers.

To perform inference on task $t$ we compute the weights of $f_t$ with $h$ then make a prediction on $x$ with $f_t$. This is performed for all tasks in parallel. During training, loss gradients are backpropagated through the target networks, then pass through the hypernetwork weights and task embeddings, such that the entire architecture can be trained directly with any gradient optimizer. Task embeddings are trained identically to other parameters.

Multi-Task Hypernetworks can naturally and effectively leverage metadata, by appending the task metadata $m_t$ the task embeddings as input to the hypernetwork $\theta_t = h(e_t, m_t)$. Unlike task embeddings, metadata are treated as constants and are frozen during training. Considering that task-specific embeddings allow Multi-Task Hypernetworks to learn task differences and relationships, by appending static metadata to the embeddings, the metadata acts as pre-learnt embedding priors from an auxiliary source. Including informative metadata may therefore improve the model's ability to learn task relationships. Further, under the hypernetwork framework, the weights that define a task's target network are a function of that task's metadata. As our model is trained with metadata over multiple tasks, the hypernetwork will directly learn the relationship between the metadata and the task functions. Our architecture is therefore able to leverage metadata in learning both task relationships and task functions explicitly.

## 2.1 Hypernetwork Architecture

The hypernetwork maps from a low-dimensional input task embedding (and optionally task metadata) to the weight matrices that define a task network. Our hypernetwork architecture (Figure 2) achieves this with two components: a feature extractor and a weight generator.

As tabular problems frequently feature available metadata, we concentrate our work on tabular problems. The hypernetwork architecture described in this section thus generates linear target networks, however we note that hypernetworks are used for efficient weight generation of various target net-

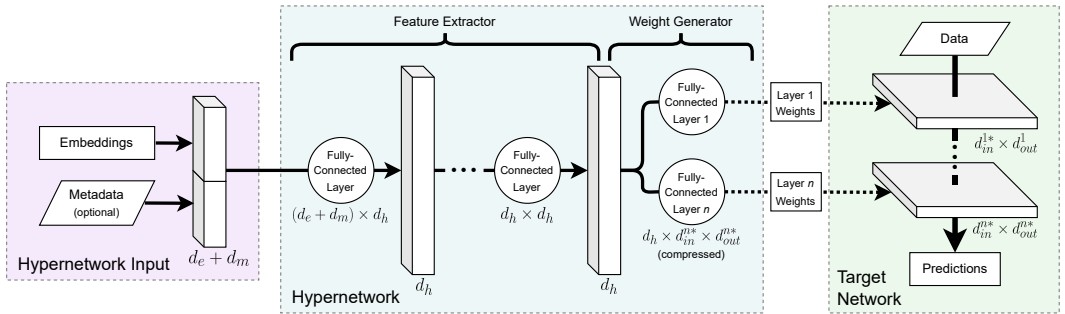

Figure 2: Multi-Task Hypernetwork architecture for a single task. The input to the hypernetwork (task embeddings and optionally task metadata) passes through the hypernetwork feature extractor layers. The resulting $d_h$ dimensional latent vector passes through the hypernetwork weight generator layers to generate the target network weights. The weight generator layers have compressed weight tensors to reduce model parameters. The target network then makes predictions for that task.

work architectures (Ha et al., 2017; Lin et al., 2020; von Oswald et al., 2020; Ye & Ren, 2021), indicating that our approach would be viable in diverse settings.

The hypernetwork takes as input a task embedding (and optionally task metadata) of dimension $d_e$ (or $d_e + d_m$). The feature extractor is a feedforward neural network with $l_h$ linear layers of $d_h$ neurons, allowing our model to learn abstract representations of task embeddings and metadata. Similarly to features in a typical neural network, metadata is normalised to unit mean and variance.

The weight generator consists of linear layers mapping from the resulting $d_h$ dimensional latent space to weight matrices which each define a layer in the target network. Let layer $n$ of the target network be a linear layer with $d_{in}^n$ inputs and $d_{out}^n$ outputs. This layer has a weight matrix of size $d_{in}^n \times d_{out}^n$ and a bias of size $d_{out}^n$, for a total of $(d_{in}^n + 1) \times d_{out}^n$ parameters. For simplicity, let $d_{in}^{n*} = d_{in}^n + 1$, such that layer $n$ has $d_{in}^{n*} \times d_{out}^n$ parameters. To generate these parameters, the hypernetwork requires a linear mapping from $d_h$ to $d_{in}^{n*} \times d_{out}^n$. This can be achieved with a hypernetwork weight matrix $W$ of size $d_h \times d_{in}^{n*} \times d_{out}^n$ and a bias $b$ of size $d_{in}^{n*} \times d_{out}^n$. However, $W$ contains $d_h$ times as many parameters as the target network layer, which is generally prohibitively large.

**Weight compression.** To reduce the number of parameters in the hypernetwork weight generator, we implement a simple weight compression technique for linear layers, analogous to the convolution compression in the original hypernetwork implementation (Ha et al., 2017). Instead of learning $W$ directly, we learn three smaller weight matrices, $W^A, W^B, W^C$, of size $d_h \times d_{in}^{n*}, d_h \times d_{out}^n, d_{in}^{n*} \times d_{out}^n$. These matrices expand into $W$ as follows: $W_{ijk} = W_{ij}^A W_{ik}^B W_{jk}^C$. This weight compression substantially reduces the total number of model parameters while remaining trainable via standard backpropagation. These three matrices learn interaction terms between two of the three dimensions of $W$: task features, target network layer input features and target network layer output features.

**Complexity.** The number of parameters in our architecture is given by:

$$P = \underbrace{(d_e + d_m + d_h(l_h - 1))(d_h + 1)}_{\text{hypernetwork feature extractor}} + \sum_{i=1}^{l_t} \underbrace{\left(2d_{in}^{i*}d_{out}^i + d_h d_{in}^{i*} + d_h d_{out}^i\right)}_{\text{hypernetwork weight generator}} + \underbrace{Td_e}_{\text{task embeddings}}$$

where $l_t$ is the number of layers in the target network. Note that a single target network has $\sum_{i=1}^{l_t} d_{in}^{i*} d_{out}^i$ total parameters, or half of the first term representing the hypernetwork weight generator parameters. Despite having more parameters than a single target network, the hypernetwork weights are shared for all tasks, such that our approach requires only $d_e$ additional parameters per task. This is substantially fewer than other soft parameter sharing MTL architectures, and in practice, our method has fewer parameters when modelling more than a few tasks.

We analyse the computational complexity of our method and other MTL architectures for larger networks in Appendix A, and find that our method is suitable for training large target networks.

## 3 EXPERIMENTS

We perform experiments with our approach and baseline methods on a set of tabular datasets with and without metadata. Baseline methods include metadata naively, as described in Section 2.

**Experimental setup.** Experiments were repeated across 50 seeds for each combination of hyperparameters. Data was partitioned into training, validation, and test sets in a 60/20/20 ratio. Performance on the validation set was used for hyperparameter selection and early stopping. Hyperparameter selection was performed independently with and without metadata. Regarding general hyperparameters, all methods used the learning rate $\eta \in \{10^p | p = -3, -3.5, -4, -4.5\}$ and random batches of size 64. All models used a network architecture of three fully connected layers of 32 neurons with ReLU activations. We empirically found that networks of this size had sufficient representation capacity to effectively model the test datasets. Regarding Multi-Task Hypernetwork hyperparameters, we tuned the number of layers in the hypernetwork feature extractor $l_h \in \{0, 1, 2\}$ where each layer had 32 neurons with ReLU activations. Task embeddings each had $d_e = 10$ parameters and were initialised with a uniform distribution and variance 1, using identical initialisations across tasks. We describe specific baseline hyperparameters in Appendix D. All experiments are implemented in PyTorch (Paszke et al., 2019) with a GeForce RTX 3080 GPU.

**Datasets.** We use four multi-task tabular regression datasets with available metadata and limited data instances for our experiments (Table 1). **Cubic** is a novel synthetic MTL dataset which we design to have a simple and complete relationship between task metadata and the task functions. Each task is a third order 1D polynomial with random coefficients, defined over $x \in [-1, 1]$, where the goal is to estimate $y$ given $x$. Polynomial coefficients are task metadata. The **Chlorophyll** dataset (Lehmann et al., 2023) aims to estimate the concentration of the chlorophyll-$a$ pigment in a lake given its water colour. Each task is a lake, and lake attributes are used as metadata. The **Algorithms** dataset (Brazdil et al., 1994) aims to estimate the performance of traditional ML algorithms on a list of datasets, given dataset summary statistics. We model each algorithm's performance on datasets as a task. Metadata is not from auxiliary sources but manually constructed from domain knowledge by naively categorising algorithms across binary labels *e.g.* "is_tree", "is_kernel". The **Robot Arm** dataset (Duka, 2014) aims to estimate robot arm joint angles given the end arm position. Each task is highly nonlinear and represents an arm with different joint lengths, which are given as task metadata. A detailed description of the data and metadata in each dataset is included in Appendix E .

Table 1: Summary of datasets used in experiments. $N$ represents total training instances, $D$ represents data dimensionality. All datasets other than Chlorophyll have equal task sizes.

| Dataset | $N$ | $T$ | Task represents | Data features | Data label | Task metadata |
|---|---|---|---|---|---|---|
| Cubic | 200 | 20 | a cubic function | x $(D = 1)$ | y $(D = 1)$ | polynomial coefficients *i.e.* a,b,c,d $(D = 4)$ |
| Chlorophyll | 796 | 88 | a lake | multispectral lake colour $(D = 16)$ | concentration of chlorophyll-$a$ $(D = 1)$ | features of lake *e.g.* depth, weather $(D = 20)$ |
| Algorithms | 422 | 24 | an ML algorithm | dataset statistics $(D = 11)$ | performance on dataset $(D = 1)$ | algorithm labels *e.g.* is_tree, is_kernel $(D = 8)$ |
| Robot Arm | 400 | 20 | a robot arm | arm end position $(D = 2)$ | arm joint angles $(D = 3)$ | arm joint lengths $(D = 3)$ |

**Baseline methods.** We include some general MTL architectures described in Section 4, avoiding domain-specific methods such as natural language processing (Sanh et al., 2019; Tay et al., 2021; Ye & Ren, 2021; Lopes et al., 2023) or computer vision (Bhattacharjee et al., 2022; Liu et al., 2019; Sun et al., 2021; Liu et al., 2022). We include: **MRN** (Multilinear Relational Networks) (Long et al., 2017), **Cross-stitch** networks (Misra et al., 2016), **Sluice** networks (Ruder et al., 2019), **DMTRL** (Deep Multi-Task Representation Learning) (Yang & Hospedales, 2017) and **MR** (Maximum Roaming) (Pascal et al., 2021). Regarding traditional approaches, **STL-naive** is a single task learning network with no task information. **STL** is a single task learning network which is given task information to learn task differences, either as metadata if available or as one-hot task embeddings otherwise. **Hard sharing** is a hard parameter sharing architecture (Caruana, 1997). Despite the use of metadata, we do not include (Zheng et al., 2019) as it is incompatible with continuous metadata and is otherwise equivalent to Hard sharing.

Table 2: Average task performance (RMSE) of our approach and baseline methods: with metadata, without metadata, and improvement in performance (reduction in RMSE) from using metadata.

|  | Method | Cubic | Chlorophyll | Algorithms | Robot Arm |
|---|---|---|---|---|---|
| | STL-naive | $0.624 \pm 0.001$ | $0.521 \pm 0.007$ | $1.950 \pm 0.062$ | $0.559 \pm 0.001$ |
| | STL | $0.132 \pm 0.007$ | $0.512 \pm 0.010$ | $1.465 \pm 0.045$ | $0.547 \pm 0.002$ |
| | Hard sharing | $\mathbf{0.040} \pm 0.002$ | $0.851 \pm 0.041$ | $1.553 \pm 0.052$ | $0.596 \pm 0.004$ |
| | MRN | $0.053 \pm 0.003$ | $0.911 \pm 0.045$ | $1.488 \pm 0.052$ | $0.565 \pm 0.003$ |
| No Metadata | Cross-stitch | $0.057 \pm 0.005$ | $0.554 \pm 0.020$ | $\mathbf{1.387} \pm 0.049$ | $0.624 \pm 0.001$ |
| | Sluice | $0.064 \pm 0.007$ | $0.534 \pm 0.018$ | $1.447 \pm 0.052$ | $0.540 \pm 0.003$ |
| | DMTRL | $\mathbf{0.038} \pm 0.001$ | $0.535 \pm 0.017$ | $\mathbf{1.342} \pm 0.044$ | $0.625 \pm 0.001$ |
| | MR | $\mathbf{0.036} \pm 0.002$ | $0.783 \pm 0.042$ | $1.505 \pm 0.051$ | $0.585 \pm 0.004$ |
| | MT Hypernet (ours) | $0.122 \pm 0.008$ | $\mathbf{0.411} \pm 0.004$ | $\mathbf{1.397} \pm 0.048$ | $\mathbf{0.485} \pm 0.002$ |
| | STL | $0.030 \pm 0.001$ | $0.489 \pm 0.008$ | $1.607 \pm 0.049$ | $\mathbf{0.469} \pm 0.002$ |
| | Hard sharing | $0.034 \pm 0.001$ | $0.973 \pm 0.045$ | $1.520 \pm 0.050$ | $0.598 \pm 0.004$ |
| | MRN | $0.041 \pm 0.003$ | $0.938 \pm 0.056$ | $1.479 \pm 0.055$ | $0.575 \pm 0.005$ |
| Metadata | Cross-stitch | $0.032 \pm 0.001$ | $0.543 \pm 0.015$ | $\mathbf{1.325} \pm 0.045$ | $0.624 \pm 0.001$ |
| | Sluice | $0.038 \pm 0.003$ | $0.559 \pm 0.020$ | $1.403 \pm 0.049$ | $0.546 \pm 0.005$ |
| | DMTRL | $0.034 \pm 0.001$ | $0.560 \pm 0.016$ | $\mathbf{1.344} \pm 0.046$ | $0.630 \pm 0.001$ |
| | MR | $0.036 \pm 0.001$ | $0.781 \pm 0.031$ | $1.489 \pm 0.048$ | $0.589 \pm 0.003$ |
| | MT Hypernet (ours) | $\mathbf{0.023} \pm 0.000$ | $\mathbf{0.395} \pm 0.004$ | $\mathbf{1.307} \pm 0.046$ | $\mathbf{0.467} \pm 0.001$ |
| | STL | $0.102 \pm 0.007$ | $0.023 \pm 0.012$ | $-0.142 \pm 0.033$ | $0.078 \pm 0.002$ |
| | Hard sharing | $0.006 \pm 0.001$ | $-0.122 \pm 0.051$ | $0.033 \pm 0.036$ | $-0.001 \pm 0.004$ |
| | MRN | $0.012 \pm 0.002$ | $-0.026 \pm 0.065$ | $0.009 \pm 0.024$ | $-0.010 \pm 0.004$ |
| Improvement | Cross-stitch | $0.024 \pm 0.005$ | $0.011 \pm 0.020$ | $0.062 \pm 0.021$ | $0.000 \pm 0.001$ |
| | Sluice | $0.027 \pm 0.006$ | $-0.026 \pm 0.018$ | $0.043 \pm 0.020$ | $-0.006 \pm 0.003$ |
| | DMTRL | $0.004 \pm 0.001$ | $-0.025 \pm 0.017$ | $-0.002 \pm 0.026$ | $-0.005 \pm 0.001$ |
| | MR | $0.001 \pm 0.001$ | $0.002 \pm 0.049$ | $0.015 \pm 0.021$ | $-0.004 \pm 0.003$ |
| | MT Hypernet (ours) | $0.099 \pm 0.008$ | $0.015 \pm 0.005$ | $0.090 \pm 0.028$ | $0.019 \pm 0.001$ |

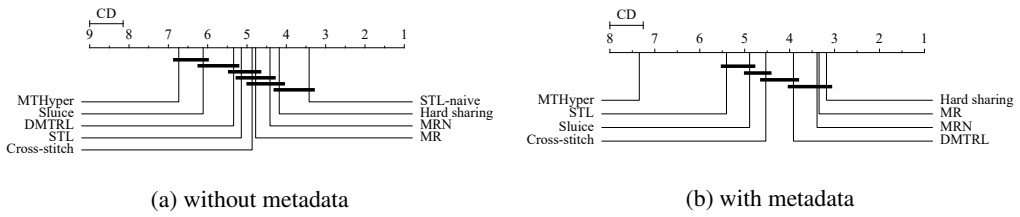

(a) without metadata          (b) with metadata

Figure 3: Critical difference diagram for Nemenyi significance test on all datasets (a) without metadata and (b) with metadata. A higher score indicates better performance.

### 3.1 EXPERIMENTAL RESULTS

**Overall performance.** Results are displayed in Table 2 as mean $\pm$ standard error. To estimate the overall difference in performance between the methods, we use a Friedman test with Nemenyi post-hoc test on the results for all datasets, displayed in Figure 3.

**Without metadata.** In the traditional MTL setting without metadata, Multi-Task Hypernetworks achieve the best performance for all datasets except Cubic. On the Robot Arm and Chlorophyll datasets, Multi-Task Hypernetworks achieve RMSE lower than other methods by 10% and 20% respectively. The Friedman-Nemenyi test shows that overall, our method outperforms all baseline methods other than Sluice networks on these datasets. These results indicate that our architecture can outperform many state-of-the-art MTL architectures on tabular problems.

**With metadata.** In the MTL setting where metadata is available during training, Multi-Task Hypernetworks achieve the best performance for all datasets. The Friedman-Nemenyi test shows that overall, our method substantially outperforms all baseline methods. Interestingly, STL outperforms all multi-task learning methods but sluice networks, despite not being designed for multi-task problems. This indicates that current MTL architectures are not able to leverage metadata effectively.

**Effects of metadata.** To analyse the impact of metadata on model performance, we consider the completeness, noisiness and complexity of the metadata. Regarding the Cubic dataset, we describe the metadata as complete, as the metadata features are sufficient to entirely represent the task functions. The metadata is noiseless as its features are generated without noise. The metadata is con-

sidered simple, since features represent coefficients which are linearly related to the task function. The positive impact of this noiseless, complete and simple metadata is reflected in the results, as metadata improves performance of all methods except MR. With available metadata, Multi-Task Hypernetworks outperform other methods by 25%.

Robot Arm metadata is also complete and generated noiselessly. Unlike cubic, Robot Arm metadata is complicated, as the relationship between the robot arm lengths and the task functions is highly nonlinear. Robot Arm metadata improves the performance of Multi-Task Hypernetworks and STL, but not other methods. This may be because although the metadata is noiseless and complete, other MTL approaches are not able to leverage this complicated metadata.

Chlorophyll metadata is noisy sensor data. It is incomplete as a lake chlorophyll-$a$ concentration has many ecological factors which cannot be entirely represented by 11 metadata features. The metadata is also complicated, as these metadata features have abstract interactions (Yang et al., 2022). Similar to Robot Arm, Chlorophyll metadata improves the performance of Multi-Task Hypernetworks and STL, but not other methods. This indicates that task knowledge can be extracted even from complex metadata, but that existing MTL techniques cannot effectively leverage it.

Algorithms metadata is manually labelled and therefore noiseless. It is simple, but incomplete as algorithm outcomes cannot be entirely represented by high-level categorical descriptions. Despite this, metadata improves the performance of Multi-Task Hypernetworks, Cross-stitch and Sluice. This demonstrates that even cheaply and naively labelled metadata can enhance model performance.

## 3.2 ABLATION STUDY

We carry out an ablation study to provide insight into the contributions of the different components in our architecture, displayed in Table 3.

Table 3: Ablation study of Multi-Task Hypernetworks with components removed (RMSE).

|  | Method | Cubic | Chlorophyll | Algorithms | Robot Arm |
|---|---|---|---|---|---|
| No Metadata | MT Hypernet | **0.122** ± 0.008 | **0.411** ± 0.004 | **1.421** ± 0.040 | **0.485** ± 0.002 |
|  | without compression | **0.103** ± 0.013 | 0.493 ± 0.007 | 1.555 ± 0.052 | 0.656 ± 0.014 |
|  | without feature extractor | **0.122** ± 0.008 | **0.413** ± 0.005 | **1.421** ± 0.040 | **0.485** ± 0.002 |
| Metadata | MT Hypernet | **0.023** ± 0.000 | **0.395** ± 0.004 | **1.322** ± 0.040 | **0.467** ± 0.001 |
|  | without compression | 0.025 ± 0.001 | 0.619 ± 0.024 | 1.935 ± 0.058 | 0.528 ± 0.003 |
|  | without feature extractor | **0.023** ± 0.000 | 0.470 ± 0.007 | **1.342** ± 0.034 | 0.474 ± 0.002 |
|  | without embeddings | 0.024 ± 0.000 | 0.434 ± 0.006 | 1.460 ± 0.046 | **0.467** ± 0.001 |
|  | with random metadata | 0.066 ± 0.004 | 0.406 ± 0.006 | 1.506 ± 0.050 | 0.517 ± 0.005 |

**Weight matrix compression reduces overparameterisation.** Using full-sized weight matrices in the hypernetwork weight generator layers without the matrix compression technique described in Section 2.1 substantially reduces performance for all datasets except Cubic without metadata. Without compression, our architecture has many more parameters and is more flexible, indicating that the weight matrix compression reduces overparameterisation and increases generalisability.

**Feature extractor learns metadata representations.** Removing the feature extractor layers from the hypernetwork decreases performance for Chlorophyll and Robot Arm with metadata. This may be because these datasets have complicated metadata, and so benefit from a non-linear feature extractor to learn abstract metadata representations. Datasets with simple metadata, or with learnt task embeddings but no metadata, may not benefit from this deep learning component. An optimal Multi-Task Hypernetwork architecture for each dataset can be found by tuning the depth of the feature extractor $l_h$, as is done in experiments in Section 3.1.

**Task embeddings provide flexibility.** Using only metadata as input to the hypernetwork without task-specific embeddings decreases performance in all datasets except Robot Arm. This may be because trainable embeddings give the hypernetwork degrees of freedom between tasks, to learn task relationships and differences more flexibly than with only static metadata. Robot Arm has complete metadata, which captures all task differences and may not benefit from additional flexibility.

**Metadata is informative.** Randomising metadata by independently shuffling each metadata feature between tasks before training the model decreases performance in all datasets. This indicates that our architecture is able to extract knowledge from informative metadata.

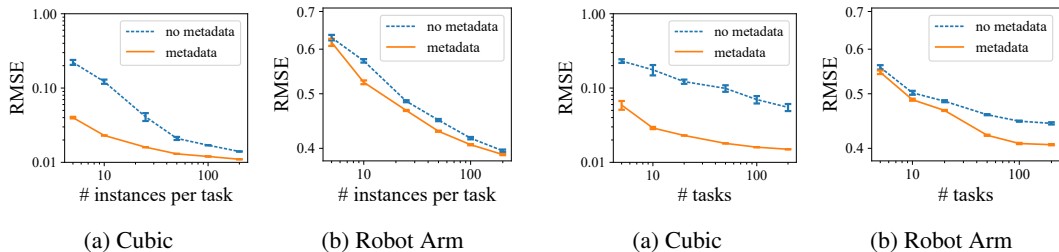

| (a) Cubic | (b) Robot Arm | (a) Cubic | (b) Robot Arm |

Figure 4: Multi-Task Hypernetwork performance vs. training instances per task

Figure 5: Multi-Task Hypernetwork performance vs. number of tasks.

**Dataset size affects metadata performance gain.** We investigate the impact of leveraging metadata on algorithmic performance for datasets of different sizes. We perform sensitivity analysis on the Cubic and Robot Arm datasets as they can be generated with arbitrary dataset sizes, varying the number of training instances in each task (Figure 4) and the number of tasks in each dataset (Figure 5). In each case, we evaluate the performance of our approach with and without metadata.

Increasing the number of training instances per task decreases the gain in performance from leveraging metadata. Intuitively, as the size of each task increases, the model is better able to learn task functions and relationships implicitly from joint optimisation without metadata. This indicates that metadata is most useful for small datasets. Increasing the number of tasks increases the gain in performance from leveraging metadata for Cubic with fewer than ten tasks, and for Robot Arm. Intuitively, as the number of tasks increases, the hypernetwork has more metadata "data points" to learn the relationship between metadata and the task functions. As Cubic has simple metadata, it may need fewer tasks to learn this relationship and hence have constant gain in performance with more than ten tasks. Robot Arm metadata is more complex, and so the performance gain from metadata increases until 100 tasks.

### 3.3 EMBEDDING ANALYSIS

We investigate whether Multi-Task Hypernetworks learn embeddings which capture meaningful task knowledge, by using embeddings to reconstruct metadata features unavailable during training. We train our model with metadata using optimal hyperparameters from experiments in Section 3.1, but completely hide one metadata feature. After training, we build a simple model which predicts the hidden metadata feature from embedding vectors learnt for each task $\{e_t\}_{t=1}^{T}$. We use a Random Forest with default parameters using scikit-learn (Pedregosa et al., 2011). We evaluate performance with 10-fold cross-validation. We repeat this process for each hidden metadata feature for each dataset over 50 seeds. Results are displayed in Table 4. To justify that metadata feature correlations do not cause compounding effects, we analyse these in Appendix B.

Table 4: Performance ($R^2$) reconstructing metadata from learnt embeddings. Displayed are the metadata feature with the best predictive performance, and the average across all metadata features.

| Performance | Cubic | Chlorophyll | Algorithms | Robot Arm |
|---|---|---|---|---|
| Best feature | $0.76 \pm 0.01$ | $0.15 \pm 0.01$ | $0.43 \pm 0.05$ | $0.53 \pm 0.02$ |
| Average | $0.59 \pm 0.01$ | $-0.14 \pm 0.00$ | $-0.04 \pm 0.02$ | $0.44 \pm 0.02$ |

Embeddings learnt on the Cubic and Robot Arm datasets display substantial explanatory power of hidden task-level information. Regarding Algorithms, some metadata features are learnt, but we find upon investigation that some features are sparse, such that there may be insufficient metadata samples to learn these accurately. Regarding Chlorophyll, the poor reconstruction performance may be explained by the noisiness, incompleteness and complexity of this metadata as discussed in Section 3.1. Overall, these results demonstrates that embeddings learnt from as few as 20 tasks can capture information which is predictive of metadata features unavailable to the model during training. This indicates that embeddings learnt by our architecture are low-dimensional representations of the task functions which contain some meaningful task-level knowledge.

## 4 RELATED WORK

**Multi-task learning architectures.** Hard sharing MTL architectures (Caruana, 1997; Wang et al., 2022; Lopes et al., 2023) have been adapted to share knowledge in task specific layers using a tensor normal distribution which learns task similarities (Long et al., 2017), to learn task differences in hard sharing layers by learning task masks (Pascal et al., 2021), to reduce generalisation error with an adversarial loss to encourage similar task latent distributions (Shui et al., 2019), and to use architecture search (Guo et al., 2020; Sun et al., 2020). However, hard sharing approaches have limited flexibility. Some soft parameter sharing architectures (Lee et al., 2018; Liu et al., 2019; Sun et al., 2021) instead transfer knowledge across tasks by linearly combining intermediate representations of each task (Misra et al., 2016; Ruder et al., 2019), or using multiple expert networks (Ma et al., 2018; Tang et al., 2020). However, these approaches are parameter-inefficient as they require learning of entire network weights for each task. Transferring knowledge by factorising model weights across tasks can improve parameter efficiency but has limited expressiveness (Yang & Hospedales, 2017). By instead transferring knowledge through a deep hypernetwork weight generator, our approach can learn abstract task relationships and is substantially more parameter-efficient than these architectures as it only learns a small embedding vector for each task.

**Multi-task learning with metadata.** Leveraging natural language task descriptions has been explored in natural language processing (Ye & Ren, 2021; Duan et al., 2021; You et al., 2016) and reinforcement learning (Sodhani et al., 2021; Wan et al., 2021). Only one prior study used metadata for MTL, by clustering tasks on their metadata and then modelling task groups with hard parameter sharing (Zheng et al., 2019). However, this unsupervised approach to learning from metadata cannot learn metadata feature importances, interactions, or relationships with task functions. Further, it is incompatible with continuous metadata and limited by its hard parameter sharing architecture.

**Hypernetworks.** Another field of neural architecture research is hypernetworks (Ha et al., 2017), introduced as a method to compress model weights, but since applied to various domains (Littwin & Wolf, 2019; Lorraine & Duvenaud, 2018; Shamsian et al., 2021; Beck et al., 2023). In transformer-based architectures, hypernetworks can condition task-specific adapters (Ye & Ren, 2021; Üstün et al., 2022; Mahabadi et al., 2021; Liu et al., 2022) and prompts (He et al., 2022) during training.

Hypernetworks can inject model-level information into deep learning systems, such as our motivation of exploiting task-level metadata. In continual learning, frozen task-specific embeddings are a memory-efficient way to store previous task models (von Oswald et al., 2020). In MTL, hypernetworks are used to incorporate user-defined task preferences and compute requirements in a hard parameter sharing tree architecture search (Raychaudhuri et al., 2022). In contrast to this, our work uses a hypernetwork as a method for soft parameter knowledge sharing between tasks.

**Task embeddings.** Task embeddings have been used to represent task relationships in continual learning (von Oswald et al., 2020) few-shot learning (Lampinen & McClelland, 2020) and meta learning (Achille et al., 2019; Lan et al., 2019). Few works learn task embeddings for MTL. In a setting where many historical models are available, Zhang et al. (2018) use task embeddings to select a model for a new MTL problem. Sun et al. (2021) use task embeddings to condition task-specific decoders; while this approach is limited to image dense prediction tasks, our work is able to condition general machine learning architectures to task embeddings by using a hypernetwork.

## 5 CONCLUSION

We propose Multi-Task Hypernetworks, a novel architecture for multi-task learning which generates weights for individual task networks with a shared hypernetwork. Task-specific embeddings produce distinct target networks from only a few parameters. Empirical results show that this architecture is effective for multi-task learning and that learnt embeddings encode meaningful task knowledge. We show that leveraging task-specific metadata is a valuable resource to improve performance on MTL problems, particularly with few data samples. Uniquely, our architecture can naturally learn tasks as a function of metadata, allowing us to leverage task-level information to learn task functions and relationships explicitly. A limitation of our work is the current lack of understanding on evaluating whether metadata will improve learning. Future research directions in multi-task learning include exploring other task embedding techniques, and investigating the potential of transfer learning to tasks with zero training instances by generating task networks from task metadata alone.

## 6 REPRODUCIBILITY

For reproducibility, we make our datasets, synthetic dataset generator and code available online at https://anonymous.4open.science/r/Multi-Task-Hypernetworks-922C.

**Datasets.** The sources for all datasets are included in Section 3. A comprehensive explanation of the data sources and generation for each dataset is included in Appendix E.

**Baselines.** A unified implementation of all baselines for linear networks, including STL, Hard sharing, MRN, Cross-stitch, Sluice networks, DMTRL and MR is included online.

**Experimental procedure.** A comprehensive description of our experimental setup is included in Section 3. Further details covering hyperparameter combinations for each baseline method are included in Section D.

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

## A COMPLEXITY ANALYSIS

In this section we investigate how the computational cost of Multi-Task Hypernetworks scales with larger target networks. We use target networks with three linear layers of sizes ranging from 32 to 4096, such that the largest of these has more parameters than ResNet-50 (He et al., 2016). We measure the average compute time and peak GPU memory allocated during a forward and backward pass over a batch with each model, displayed in Figure 6. All models were trained on Cubic with 10 tasks. The hyperparameter combination with lowest compute time was selected for each model, as there was no substantial difference between hyperparameters for any model.

Hard parameter sharing approaches are substantially faster than soft parameter sharing approaches. All soft sharing approaches follow a similar time complexity trend, and all approaches excluding DMTRL follow a similar memory complexity trend. Despite being classified as hard sharing, MR has complexity similar to soft sharing approaches, likely due to the use of task-wise model masks. We observe that our architecture has similar computational cost to other models which are commonly used to train large networks (Misra et al., 2016; Yang & Hospedales, 2017; Ruder et al., 2019; Pascal et al., 2021). We conclude that our architecture is computationally capable of scaling to applications with larger network architectures.

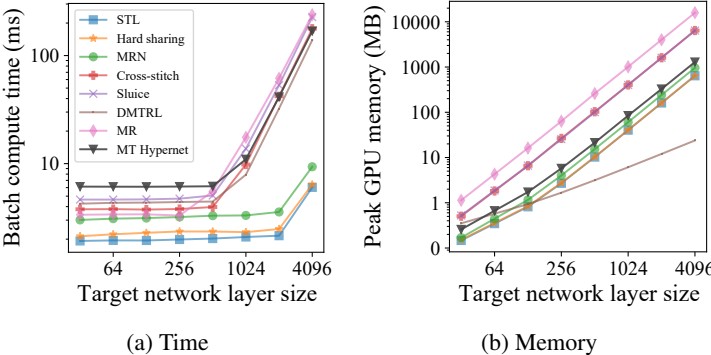

|  (a) Time | (b) Memory |

Figure 6: Model complexity vs. Target network size.

## B EMBEDDING ANALYSIS

In this section, we expand upon the experiments in Section 3.3. Specifically, we display the predictive performance of the learnt embeddings on the hidden task metadata features across all metadata features for all datasets (Figure 5), and justify that metadata feature correlations do not cause compounding effects in our tests (Figure 7).

We observe that simpler metadata features, such as *d* in Cubic which represents the constant polynomial term, tend to be predicted more accurately. Excluding *depth*, Chlorophyll metadata features are not captured by the learnt embeddings. This may be explained by the noise, incompleteness and high complexity of this metadata discussed in Section 3.1.

**Metadata feature correlations.** We investigate whether correlated metadata features may be impacting the performance of metadata feature reconstruction from embeddings. Specifically, the learnt embeddings may be affected by the remaining metadata features, which are available during training of the Multi-Task Hypernetwork and may be correlated to the hidden feature. We emphasise that these metadata features are not available to the Random Forest model which reconstructs the missing metadata features from the learnt embeddings.

We compute the Kendall rank correlation of metadata features, displayed in Figure 7. We find that Cubic metadata has no strong correlations between metadata features. This indicates that the predictive capacity of the learnt embeddings is not affected by other metadata features. Robot Arm metadata has some correlated features, specifically *length_2* and *length_3*, although we note that these are generated as independent random variables so this correlation is spurious. Although Algorithm has correlations between *is_tree* and *is_inductive_tree*, and *is_non_para* and *is_bayes*, metadata

Table 5: Performance ($R^2$) of reconstructing missing metadata features from learnt embeddings, for all metadata features for all datasets.

| Dataset | Metadata Feature | | | |
|---|---|---|---|---|
| Cubic | $a$
$0.35 \pm 0.02$ | $b$
$0.72 \pm 0.02$ | $c$
$0.54 \pm 0.03$ | $d$
**0.76** $\pm 0.01$ |
| Chlorophyll | *depth*
**0.15** $\pm 0.01$
*area*
-0.08 $\pm 0.01$
*laketype_1*
-0.23 $\pm 0.02$ | *water_temp*
-0.09 $\pm 0.01$
*%_cropland*
-0.15 $\pm 0.01$
*laketype_2*
-0.23 $\pm 0.02$ | *elevation*
-0.23 $\pm 0.01$
*%_pasture*
-0.27 $\pm 0.02$
*laketype_3*
-0.33 $\pm 0.02$ | *latitude*
-0.09 $\pm 0.02$
*rainfall*
-0.21 $\pm 0.02$ |
| Robotarm | *length_1*
0.33 $\pm 0.03$ | *length_2*
0.46 $\pm 0.03$ | *length_3*
**0.53** $\pm 0.02$ | |
| Algorithms | *is_kernel*
**0.43** $\pm 0.05$
*is_non_para*
0.02 $\pm 0.02$ | *is_inductive_tree*
-0.31 $\pm 0.04$
*is_classic_stats*
0.31 $\pm 0.05$ | *is_rule_based*
0.01 $\pm 0.03$
*is_bayes*
-0.49 $\pm 0.02$ | *is_NN*
-0.27 $\pm 0.03$
*is_tree*
0.01 $\pm 0.03$ |

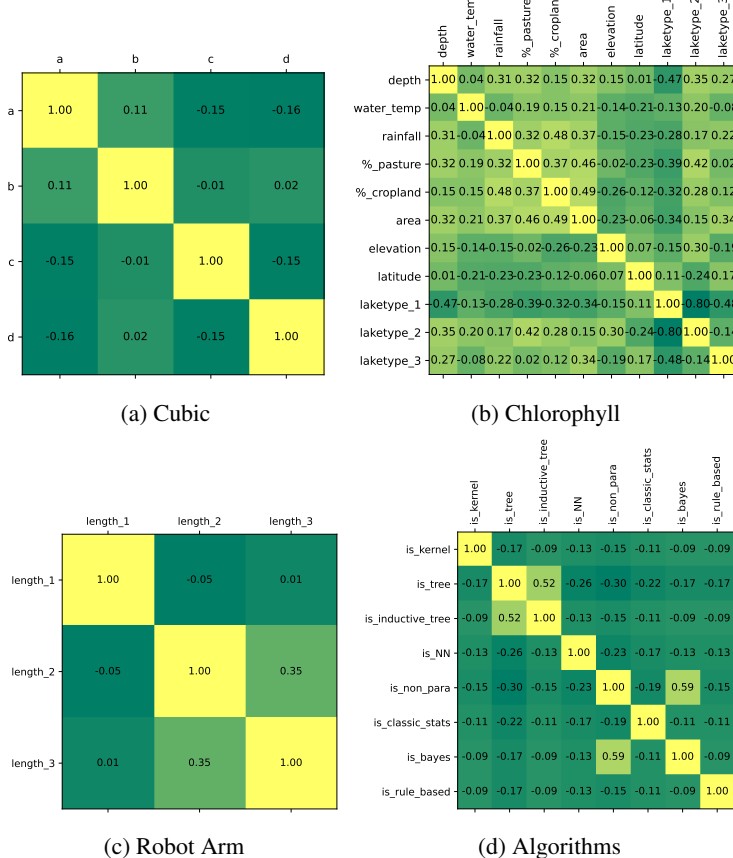

(a) Cubic  (b) Chlorophyll

(c) Robot Arm  (d) Algorithms

Figure 7: Kendall rank correlation coefficient of metadata features for each dataset.

reconstruction performance was greatest for *is_kernel* and *is_classic_stats*, which have no strong metadata correlations. Chlorophyll has many highly correlated metadata features, but these are inconsequential as embeddings did not have substantial explanatory power on any of these features. Pearson correlation coefficient and Spearman correlation coefficients produced similar results.

We conclude that the predictive capacity of the learnt embeddings is not affected by metadata features available during training for any dataset, as these are not strongly correlated to the metadata features which were accurately captured by the embeddings.

## C  HYPERNETWORK INITIALISATION

In this section, we discuss the parameter initialisation scheme used by our hypernetwork architecture. Default neural network parameter initialisation schemes such as Kaiming initialisation (He et al., 2015) are motivated by the assumption of fixed variance of neuron activation through network layers. One work on hypernetworks by Chang et al. (2019) show that these assumptions do not hold for hypernetwork architectures, and instead propose their own initialisation schemes. In our work, we experimented with the initialisation of the shared hypernetwork weights using both of these initialisation schemes and found no significant difference in model performance. For simplicity, we used Kaiming initialisation in our experiments.

## D  BASELINE HYPERPARAMETERS

We list the hyperparameters tuned for each baseline method in our experiments (Table 6). Note that STL-naive, STL, and Hard sharing are traditional neural network architectures with no algorithm-specific hyperparameters. For Cross-stitch and Sluice networks, $\alpha_{init}$ and $\beta_{init}$ refer to the initialisation schemes for the $\alpha$ and $\beta$ model parameters respectively. For MR, $p$ refers to the fraction of neurons used for each task mask, and $\Delta$ refers to the epoch frequency of mask changes. # Combinations represents the total number of hyperparameter combinations tested during tuning, where each combination was tested with 50 seeds.

**Network capacity.** All methods used the same network capacity: three layers of 32 neurons. Since MR uses only a subset of the network for each task, the number of neurons in the network was increased according to $p$ such that the capacity of each task would be of three layers of 32 neurons.

**Early stopping.** All methods calculated validation loss only every 5 epochs for computation efficiency. Early stopping was triggered if the validation loss increased over 20 epochs.

Table 6: Hyperparameters used in experiments for baseline methods.

| Method | Fixed Parameters | Tuned Parameters | # Combinations |
|---|---|---|---|
| All methods | | $\eta \in \{10^p \mid p = -3, -3.5, -4, -4.5\}$ | |
| STL-naive | | | 4 |
| STL | | | 4 |
| Hard sharing | | | 4 |
| MRN | | $\lambda_{\text{tradeoff}} \in \{10^p \mid p = -2, -3, -4, -5\}$, $k \in \{1, 0.1, 0.01\}$ | 48 |
| Cross-stitch | $\alpha_{\text{init}} = $ "balanced" | | 4 |
| Sluice | $n_{\text{subspace}} = 2$, $\beta_{\text{init}} = $ "imbalanced" | $\alpha_{\text{init}} \in \{$"balanced", "imbalanced"$\}$, $\lambda_{\text{orth}} \in \{10^p \mid p = 0, -1, -2, -3, -4\}$ | 40 |
| DMTRL | | method $\in \{$"Tucker", "TT"$\}$, $k \in \{2, 4, 8, 16\}$ | 32 |
| MR | | $p \in \{0.4, 0.6, 0.8\}, \Delta \in \{1, 2, 4, 8\}$ | 48 |
| MT Hypernet | $d_e = 10$ | $l_h \in \{1, 2, 3\}$ | 12 |

## E  DATASETS

In this section, we expand upon the dataset descriptions given in Section 3, including examples of dataset metadata, and a comprehensive explanation of the data sources and generation for each dataset.

**Examples of Metadata.** We include example metadata features from the test datasets included in this paper, displayed in Table 7, Table 8, Table 9 and Table 10. We include metadata from four tasks for each dataset. All datasets are also available in full online.

**Cubic.** The coefficients for the cubic polynomials which serve as tasks in the Cubic dataset are generated from a uniform distribution $a, b, c, d \sim \mathcal{U}[-1, 1]$. The only feature, $x$, is uniformly generated from the same domain, and the label $y$ is computed as $y = ax^3 + bx^2 + cx + d$.

Table 7: Example metadata for Cubic dataset.

| Cubic | a | b | c | d |
|---|---|---|---|---|
| $0.1x^3 + 0.6x^2 + 0.4x + 0.2$ | 0.106 | 0.580 | 0.366 | 0.225 |
| $0.4x^3 + 0.7x^2 - 0.4x + 0.4$ | 0.388 | 0.667 | -0.429 | 0.368 |
| $-0.9x^3 - 0x^2 + x + 0.8$ | -0.863 | -0.029 | 0.962 | 0.809 |
| $-0.4x^3 - 0.8x^2 - 0.9x + 0.9$ | -0.384 | -0.760 | -0.872 | 0.862 |

**Chlorophyll.** The metadata for the Chlorophyll dataset was obtained from the LakesATLAS dataset (Lehner et al., 2022) which contains relevant lake attributes. This auxiliary data was joined to the GLORIA dataset (Lehmann et al., 2023) by matching the datasets at the lake level. Lakes with less than 5 data points were discarded, as this was insufficient to model these tasks.

Table 8: Example metadata for Chlorophyll dataset.

| Lake | elevation | water_temp | area_log | depth_log | precipitation | water_type_0 | water_type_1 |
|---|---|---|---|---|---|---|---|
| Lake Erie | 180.796 | 12.454 | 11.115 | 3.718 | 7.349 | 0 | 0 |
| Lake Taihu | 3.3 | 16.012 | 7.753 | 0.833 | 7.05 | 0 | 0 |
| Lake Kasumigaura | 142.705 | 7.558 | 5.126 | 1.96 | 7.287 | 0 | 0 |
| Eagle Creek Reservoir | 238 | 21.227 | -1.204 | 1.163 | 6.14 | 1 | 0 |

**Algorithms.** We designed simple binary metadata features for the Algorithms dataset (Brazdil et al., 1994) by creating categories that separated algorithms into different classes. This was done naively without robust evaluation. We note that some Algorithm metadata features are sparse, which may be insufficient for learning for the embedding analysis experiments as described in 3.3.

Table 9: Example metadata for Algorithms dataset.

| Algorithm | is_kernel | is_tree | is_inductive_tree | is_NN | is_non_para | is_classic_stats | is_bayesian | is_rule_classifier |
|---|---|---|---|---|---|---|---|---|
| Ac2 | 0 | 1 | 0 | 0 | 0 | 0 | 0 | 0 |
| IndCART | 0 | 1 | 0 | 0 | 0 | 0 | 0 | 0 |
| KNN | 1 | 0 | 0 | 0 | 0 | 0 | 0 | 0 |
| Kohonen | 0 | 0 | 0 | 1 | 0 | 0 | 0 | 0 |

**Robot Arm.** A two-dimensional, three-arm version of the Robot Arm dataset (Duka, 2014) was used in our experiments. This produces an ill-posed function with three labels, which is highly non-linear and therefore difficult to learn. Robot arm lengths are generated with a uniform distribution *length_1, length_2, length_3* $\sim \mathcal{U}(0, 1]$, and the arm angles are distributed $\theta_1, \theta_2 \sim \mathcal{U}[-\pi, 0]$, $\theta_3 \sim \mathcal{U}[-\pi/2, \pi/2]$. The end arm position $(y_1, y_2)$ is computed from these parameters using trigonometric equations described in (Duka, 2014)

Table 10: Example metadata for Robot Arm dataset.

| Robot_Arm_ID | joint_length_1 | joint_length_2 | joint_length_3 |
|---|---|---|---|
| 0 | 0.549 | 0.715 | 0.603 |
| 1 | 0.058 | 0.434 | 0.312 |
| 2 | 0.696 | 0.378 | 0.180 |
| 3 | 0.490 | 0.227 | 0.254 |

## F    RELATED WORK COMPARISON

In this section we illustrate the differences between our approach and existing works described in Section 4. We display this in Table 11.

Regarding the referenced algorithms, Hard sharing approaches include the original MTL architecture (Caruana, 1997), MRN (Long et al., 2017), MR (Pascal et al., 2021), AMTNN (Shui et al., 2019). Soft sharing approaches include Cross-stitch (Misra et al., 2016), Sluice (Ruder et al., 2019), and MMoE (Ma et al., 2018). Specific algorithms included are DMTRL (Yang & Hospedales, 2017),

Table 11: Overview of related MTL architectures.

| MTL architectures | Flexible | Parameter-efficient | Task embeddings | Hypernetwork | Tabular data | Metadata |
|---|---|---|---|---|---|---|
| Hard sharing approaches | ✗ | ✓ | ✗ | ✗ | ✓ | ✗ |
| Soft sharing approaches | ✓ | ✗ | ✗ | ✗ | ✓ | ✗ |
| DMTRL | ✓ | ✓ | ✗ | ✗ | ✓ | ✗ |
| TSN | ✗ | ✓ | ✓ | ✗ | ✗ | ✗ |
| mCMTL | ✗ | ✓ | ✗ | ✗ | ✓ | ✓ |
| Transformer hypernetworks | ✗ | ✓ | ✗ | ✓ | ✗ | ✓ |
| MT Hypernet (ours) | ✓ | ✓ | ✓ | ✓ | ✓ | ✓ |

TSN (Sun et al., 2021) and mCMTL (Zheng et al., 2019). Transformer hypernetworks include various works (Ye & Ren, 2021; Üstün et al., 2022; He et al., 2022; Mahabadi et al., 2021; Liu et al., 2022)

Regarding the columns, **Flexible** indicates approaches which can learn task differences at every layer of the task networks. **Parameter-efficient** indicates approaches which do not have to learn weights for an entire new target network for each task. **Task embeddings** indicates approaches that represent tasks by learning a task-specific embedding. **Hypernetwork** indicates techniques that use a hypernetwork component. **Tabular data** indicates approaches that are suitable for modelling tabular data such as the datasets used for experiments in this paper. **Metadata** indicates approaches that can effectively leverage task-level metadata in the learning process. This illustrates that our proposed architecture is substantially different to existing works.

