# OpenReview forum: "Multi-Task Learning with Hypernetworks and Task Metadata"
_ICLR.cc/2024/Conference — ICLR 2024 Conference Withdrawn Submission_

### Official Review · Reviewer_66Fq · 2023-10-16

**Soundness:** 2 fair
**Presentation:** 3 good
**Contribution:** 2 fair
**Rating:** 5
**Confidence:** 4

**Summary:**

This paper proposed a multi-task architecture for tabular data. The proposed architecture includes a hypernetwork which is used to generate model weights for different tasks from task-specific embedding and metadata. The resulting multi-task model is a soft-parameter sharing network that can get good accuracy on datasets with metadata.

**Strengths:**

- The paper is easy to follow and the proposed method is easy to understand.
- The experiments show the effectiveness of the proposed method especially the usage of meta in tabular data.

**Weaknesses:**

- Motivation and Practical Usage:
  - Limited Applicability: One major issue pertains to the paper's motivation and practical usage. The proposed method is primarily suited for tabular data, severely restricting its versatility due to its reliance on metadata.
  - Limited Advantages: Furthermore, the paper falls short in demonstrating notable advantages. It does not address key considerations, such as storage or computational efficiency. The motivation behind the trade-off between parameter efficiency and task accuracy, as mentioned in the abstract, is insufficiently explored. In many Multi-Task Learning (MTL) scenarios, like those involving multiple vision tasks in autonomous driving, one expects to reduce storage costs through parameter sharing and even lower computational costs through computation sharing. Regrettably, the proposed method offers no benefits in these two aspects, as it relies on soft-parameter sharing.

- Novelty:
  - Lack of Novelty: Another significant concern pertains to the novelty of the proposed approach. The paper introduces hypernetworks as a central component, which is not a novel concept in MTL. Hypernetworks have already been extensively explored in both Natural Language Processing (NLP) [1] and Vision MTL scenarios [2], diminishing the originality of the paper's contribution.
  - Common Practice: Moreover, employing task embedding as input is a conventional practice in the field of MTL. This conventional approach further diminishes the uniqueness of the proposed method.
  - Limited Generalization: While the paper introduces an insightful use of metadata, its applicability is severely constrained. This novel aspect can only be employed effectively in the context of tabular data, limiting its broader relevance and potentially affecting its acceptance at top-tier conferences.

[1] Mahabadi, Rabeeh Karimi, et al. "Parameter-efficient multi-task fine-tuning for transformers via shared hypernetworks." arXiv preprint arXiv:2106.04489 (2021).
[2] Liu, Yen-Cheng, et al. "Polyhistor: Parameter-efficient multi-task adaptation for dense vision tasks." Advances in Neural Information Processing Systems 35 (2022): 36889-36901.

**Questions:**

- What do you think makes your method different from the existing methods? In other words, what novelty would you like to emphasize in your method?
- Do you think it is possible to use metadata in general multi-task problems like vision tasks and NLP tasks?

---

> ### Author Response · Authors · 2023-11-13
> **R4: Response to 66Fq**
>
> We earnestly thank the reviewer for their time in reading our paper and leaving a detailed and equitable review.
> * Regarding storage cost by parameter sharing, our method does offer substantial advantages - namely, substantially reduced marginal cost of adding tasks. For each additional task added to the model, only d_e additional parameters are required. In our experiments, we use only 10 parameters per task.
> Q1) Novelty we wish to emphasise:
> * Our method is substantially more parameter-efficient than existing soft parameter sharing approaches. Note that when we discuss soft parameter sharing approaches, we are using this term to describe architectures which can generate entirely distinct weights for each task network. Our method achieves this, despite the fact that most learnable model parameters are in the hypernetwork, which is shared between all tasks.
> * Our method is the first method to leverage task metadata for general (non-NLP) MTL problems.
> Q2) It is certainly possible to use metadata in general multi-task problems like vision tasks and NLP tasks. However, regarding vision problems, datasets tend to not have existing metadata. Regarding NLP tasks, metadata exists and has been exploited in existing works such as [1], however these methods rely on NLP-specific architectures.
>
> [1] Qinyuan Ye and Xiang Ren. Learning to generate task-specific adapters from task description. In Proceedings of the 59th Annual Meeting of the Association for Computational Linguistics and the 11th International Joint Conference on Natural Language Processing, pp. 646–653, 2021.

---

### Official Review · Reviewer_bXYN · 2023-10-17

**Soundness:** 2 fair
**Presentation:** 4 excellent
**Contribution:** 2 fair
**Rating:** 5
**Confidence:** 4

**Summary:**

This article presents a method for multi-task learning using hypernetworks, specifically designed for small tabular datasets by leveraging their metadata. A common hypernetwork produces network weights derived from both learned embeddings and the task's metadata. These weights are tailored for each task and are utilized to produce the final predictions relevant to that task.

**Strengths:**

- The article is well-composed and systematically arranged.

- The concept of employing hypernetworks to produce weights specific to each task is intriguing, particularly with the incorporation of supplementary data (referred to as metadata in the paper). This approach holds potential for crafting robust multi-task networks.

- The assessments are thorough, encompassing many statistical nuances.

**Weaknesses:**

- The evaluation size for the multi-task learning approach seems limited, with fewer than 1,000 training samples. Determining the efficacy of the suggested deep learning technique from such a compact dataset can be challenging. Other multi-task learning studies have set more expansive benchmarks such as PASCAL-Context, NYUD, and Cityscapes [1],

- While the paper touches upon some older deep learning techniques (like Cross-stitch and Sluice), it misses out on discussing or comparing several recent deep learning-based multi-task learning methodologies. Notable omissions include InvPT [1], TaskPrompter [2], TaskExpert [3], MQTransformer [4], and MTFormer [5]. Notably, TaskExpert [3] adopts a strategy of learning task-specific gating networks for generating task-dependent weights during task-specific decoding, which bears a resemblance to the hypernetwork learning process to some extent.

- The parameter size and computational efficiency of the methods listed in Table 2 are not shown. I understand that Fig. 6 provides some information about efficiency, but we would like to see the effectiveness vs. efficiency.

References:

[1] Inverted Pyramid Multi-task Transformer for Dense Scene Understanding. ECCV 2022

[2] TaskPrompter: Spatial-Channel Multi-Task Prompting for Dense Scene Understanding. ICLR 2023

[3] TaskExpert: Dynamically Assembling Multi-Task Representations with Memorial Mixture-of-Experts. ICCV 2023

[4] Multi-Task Learning with Multi-Query Transformer for Dense Prediction. arXiv 2022

[5] MTFormer: Multi-Task Learning via Transformer and Cross-Task Reasoning. ECCV 2022

**Questions:**

- Can you provide a rationale for evaluating multi-task models using just a single tiny-scale multi-task benchmark?

- How does the performance stack up against one or two of the aforementioned cutting-edge multi-task architectures?

- How does the model perform on datasets such as PASCAL-Context?

I would be open to revising my final score after reviewing the authors' response. Thank you.

---

> ### Author Response · Authors · 2023-11-13
> **R3: Response to bXYN**
>
> We sincerely thank the reviewer for reading our paper and leaving a well-supported review.
> Q1) The reason we don't explore these vision datasets in our paper is that these datasets do not have existing metadata - in fact, most vision problems do not, which is why we focus on tabular problems. Given the NYUv2 dataset, for example, the tasks are depth estimation and semantic segmentation. It is not clear how to describe the task of depth estimation with metadata features.
> We also note that although we focus on small datasets, we include experiments in the appendix which demonstrate that our method is computationally suitable for training larger backbone models.
> Q2) We thank the reviewer for their list of relevant papers, particularly TaskExpert. However, we note that these approaches and others referenced in our paper such as [1] are only suited to vision tasks, and cannot easily be applied to other MTL architectures.
> We appreciate the suggestion of tabulating baseline algorithm efficiency, which we intend to incorporate in future versions of the paper.
>
> [1] Task switching network for multi-task learning. CVF 2021

---

### Official Review · Reviewer_88C4 · 2023-10-20

**Soundness:** 3 good
**Presentation:** 3 good
**Contribution:** 2 fair
**Rating:** 3
**Confidence:** 4

**Summary:**

This paper proposes a hypernetwork to generate flexible task networks for each task from different task-specific embeddings and metadata. They show empirically that the proposed method outperforms many MTL architectures on small tabular data problems, and leverage metadata more effectively than existing methods.

**Strengths:**

1. The motivation of this paper is clear.
2. The proposed method can generate flexible task networks.

**Weaknesses:**

1. The novelty of this paper is limited. The ideas of utilizing hypernetworks for MTL and learning task-specific embeddings have been extensively employed in existing MTL research. The proposed method of this paper is relatively straightforward and does not provide significant novel insights, just an application work not a research paper.
2. The authors claim that "We additionally show experimentally that the task embeddings learn “meaningful” task representations, in that they are predictive of task-level knowledge." However, Section 3.3 only shows the task embeddings can be used to reconstruct metadata and is based on a  model that is needed to be learned. How can this demonstrates the meaningfulness of learned task embeddings? Furthermore, the reconstructed performance is poor.
3. The authors only conducts experiments on small-scale datasets, with the maximum dataset size being 796. It seems challenging to generalize their method to address large-scale MTL tasks. This is because tackling large-scale MTL problems typically requires a target network model with a significantly larger parameter set, which becomes difficult to predict using the hypernetwork. Although the authors propose a method to reduce the dimension of parameters in the hypernetwork generator using an existing technique, it remains unclear whether this compression technique would successfully scale up the hypernetwork to effectively handle large-scale real-life MTL problems.
4. The authors claim on the superior performance of their method paper, by saying the proposed method outperforms SOTA methods. However, the compared methods in this paper are not SOTA methods.  More recent SOTA methods are needed to discuss and compare.

**Questions:**

see Weakness

---

> ### Author Response · Authors · 2023-11-13
> **R2: Response to 88C4**
>
> We graciously thank the paper for their time in reading our paper and for this review.
> 1. Novel components of the proposed method are (A) MTL architecture where task-specific weights are a function of a task-specific embedding, split into a feature extractor and weight generator and (B) design of a hypernetwork architecture for linear target models, including a weight compression scheme.
> 2. The metadata of a given task is assumed to be related to the task function in some way. As we are able to reconstruct the metadata from the learnt embedding, we are demonstrating that the learnt embedding is capturing a representation of the task function, and hence describe it as "meaningful". When considering the reconstruction performance, it is important to note that this reconstruction is performed with few samples (as few as 20 tasks to learn embedding representations), and in some cases predicts metadata which is only abstractly and incompletely related to the task function.
> 3. We include complexity analysis experiments in Appendix A, which show that our method is computationally suitable to large target networks.
> 4. We describe the methods in the paper as SOTA methods as many more recent methods are not applicable to the datasets included in the paper. However, we agree with the reviewer that this terminology can be misleading, and will change this in future versions of the paper.

---

### Official Review · Reviewer_RnjU · 2023-11-01

**Soundness:** 2 fair
**Presentation:** 3 good
**Contribution:** 2 fair
**Rating:** 3
**Confidence:** 3

**Summary:**

The paper presents Multi-Task Hypernetworks that aim to leverage the information from different tasks to improve the performance on each task. In particular, hypernetworks with weight compression are employed in this work to allow knowledge transfer across tasks while introducing a minimal number of trainable parameters. Furthermore, the authors propose to utilize the metadata, which could help during training. Experimental results on different tabular datasets validate the effectiveness of the proposed method.

**Strengths:**

- The paper is well-written and easy to follow. The authors present their method in detail.
- The proposed method outperforms other baselines on different experimental setups. Ablation studies are conducted thoroughly to understand the effectiveness of each component.

**Weaknesses:**

- The proposed method is a simple combination of employing hypernetwork and more informative inputs (using handcrafted metadata). Moreover, the idea of utilizing HyperNetwork for Multi-task learning is not novel. It has been studied in prior work (e.g. [1,2]).
- The construction of metadata requires domain knowledge from ML practitioners and potentially includes target leakage features.
- The number of trainable parameters in the proposed method is larger than the target network, which is inefficient for training. This is addressed by using chunk embeddings for different small parts of the main network.

[1] Navon, Aviv, et al. "Learning the Pareto Front with Hypernetworks." International Conference on Learning Representations. 2020.

[2] Lin, Xi, et al. "Controllable pareto multi-task learning." arXiv preprint arXiv:2010.06313 (2020).

[3] von Oswald, Johannes, et al. "Continual learning with hypernetworks." International Conference on Learning Representations. 2019.

**Questions:**

- Please add the number of trainable parameters/training budgets used for comparative methods in the main tables.
- Can you compare your proposed hypernetwork with weight compression against the chunking technique?

---

> ### Author Response · Authors · 2023-11-14
> **R1: Response to RnjU**
>
> We sincerely thank the reviewer for their time reading our paper and for their careful review.
> * Although the idea of employing informative inputs as hypernetwork embeddings is not novel, the idea of embedding task-specific auxiliary information in a hypernetwork has not been explored outside of the field of NLP. Likewise, hypernetworks have been used in multi-task learning, but in a different manner and for a different purpose. The works [1,2] use a hypernetwork to generate the weights of an entire MTL (generally hard sharing, shared-bottom) architecture, based on some model-level input. In contrast, our work uses a hypernetwork to generate the weights of the network for a single task network. Conceptually, their work uses a hypernetwork to condition the entire model to some set of conditions, while our work uses a hypernetwork as the system for soft parameter sharing between tasks and conditioning task models to task-specific metadata.
> * Metadata is defined at the task level, and is hence constant across the training, validation and test set. There is therefore no chance of leakage from the test set.
> * The number of parameters in the proposed method is larger than a single target network. However, as it requires only d_e additional parameters per extra task, the method has substantially fewer parameters than existing soft parameter approaches when modelling more than a few tasks.
> Q1) We appreciate the suggestion of tabulating baseline algorithm parameter counts, which we intend to incorporate in future versions of the paper.
> Q2) It is not clear how to adapt the chunking technique used in the original hypernetwork paper from convolutional layers to linear layers. This is the motivation for the development of our own compression technique described in the paper.

---

### Author Response · Authors · 2023-11-14
**R0: Responses to comments made by multiple reviewers:**

**Experiments on different ML problems (vision problems or NLP).** The reason we do not perform experiments on vision problems including popular datasets (Cityscapes, NYUv2, etc.) in our paper is that these datasets do not have task metadata. Given the NYUv2 dataset, for example, the tasks are depth estimation and semantic segmentation. It is not clear how to describe the task of depth estimation with metadata features. Regarding NLP tasks, metadata exists and has been exploited in existing works [1], however these methods rely on NLP-specific architectures. In contrast, many multi-task tabular data problems exist with available metadata, and hence we focus our research on these problems.

**More SOTA baselines.** As most recent MTL research is carried out on vision or NLP problems, we find that the majority of SOTA algorithms are specifically designed for these settings. The baselines we include are recent works which were tested on vision problems but contain no vision-specific components, and can therefore be used to model tabular problems.

**Algorithm efficiency for larger models.** We include complexity analysis experiments in Appendix A, which show that our method is computationally suitable to large target networks.

[1] Qinyuan Ye and Xiang Ren. Learning to generate task-specific adapters from task description. In Proceedings of the 59th Annual Meeting of the Association for Computational Linguistics and the 11th International Joint Conference on Natural Language Processing, pp. 646–653, 2021.